# Neurofibromatosis Type 1: Pediatric Aspects and Review of Genotype–Phenotype Correlations

**DOI:** 10.3390/cancers15041217

**Published:** 2023-02-14

**Authors:** Cristina Peduto, Mariateresa Zanobio, Vincenzo Nigro, Silverio Perrotta, Giulio Piluso, Claudia Santoro

**Affiliations:** 1Department of Precision Medicine, University of Campania “Luigi Vanvitelli”, Via Luigi de Crecchio 7, 80138 Naples, Italy; 2Telethon Institute of Genetics and Medicine (TIGEM), Via Campi Flegrei 34, 80078 Pozzuoli, Italy; 3Department of Women’s and Children’s Health and General and Specialized Surgery, University of Campania “Luigi Vanvitelli”, Via Luigi de Crecchio 2, 80138 Naples, Italy; 4Clinic of Child and Adolescent Neuropsychiatry, Department of Physical and Mental Health, and Preventive Medicine, University of Campania “Luigi Vanvitelli”, Largo Madonna delle Grazie 1, 80138 Naples, Italy

**Keywords:** neurofibromatosis type 1 (NF1), genotype–phenotype correlations, pediatric features

## Abstract

**Simple Summary:**

In the last few years, an increasing number of genotype–phenotype correlations has been described for neurofibromatosis type 1 (NF1), impacting on the clinical follow-up of patients, especially in pediatric age. The widespread use of molecular diagnosis, made easier by next generation sequencing technology, now allows very early confirmation of clinical diagnosis, even in the case of non-canonical presentation of the disorder with other overlapping conditions. Here, we review the main clinical characteristics and complications related to NF1, particularly those occurring in children. We also describe currently known genotype–phenotype associations that need to be considered because of their effect on genetic counseling and prognosis. Molecular diagnosis is today fundamental for the confirmation of doubtful clinical diagnoses, especially in the light of recently revised diagnostic criteria, and for the early identification of genotypes, albeit few, that correlate with specific phenotypes.

**Abstract:**

Neurofibromatosis type 1 (NF1) is an autosomal dominant condition, with a birth incidence of approximately 1:2000–3000, caused by germline pathogenic variants in *NF1*, a tumor suppressor gene encoding neurofibromin, a negative regulator of the RAS/MAPK pathway. This explains why NF1 is included in the group of RASopathies and shares several clinical features with Noonan syndrome. Here, we describe the main clinical characteristics and complications associated with NF1, particularly those occurring in pediatric age. NF1 has complete penetrance and shows wide inter- and intrafamilial phenotypic variability and age-dependent appearance of manifestations. Clinical presentation and history of NF1 are multisystemic and highly unpredictable, especially in the first years of life when penetrance is still incomplete. In this scenario of extreme phenotypic variability, some genotype–phenotype associations need to be taken into consideration, as they strongly impact on genetic counseling and prognostication of the disease. We provide a synthetic review, based on the most recent literature data, of all known genotype–phenotype correlations from a genetic and clinical perspective. Molecular diagnosis is fundamental for the confirmation of doubtful clinical diagnoses, especially in the light of recently revised diagnostic criteria, and for the early identification of genotypes, albeit few, that correlate with specific phenotypes.

## 1. Background

Neurofibromatosis type 1 (NF1; MIM 162200) is an autosomal dominant condition with complete penetrance and variable expressivity. The incidence at birth is 1:2000–3000 individuals among the general population [1,2,3,4]. In about 50% of cases, it is caused by de novo mutations in *NF1* on chromosome 17q11.2; *NF1* encodes neurofibromin, a ubiquitous protein mainly expressed in neurons, Schwann cells, and glial cells. The condition is today included in the group of RASopathies due to neurofibromin-mediated inactivation of the RAS/MAPK pathway (Figure 1), with secondary regulation of proliferation, differentiation, cell migration, and apoptosis [5,6]. RASopathies also include Noonan (MIM 163950), LEOPARD (MIM 151100), Costello (MIM 218040), Cardio-facio-cutaneous (MIM 115150), and Legius (LGSS; MIM 611431) syndromes. All these disorders share several phenotypic characteristics (e.g., skin involvement, neurocognitive impairment, dysmorphisms, increased oncological risk), in some cases making differential clinical diagnosis complex [7,8,9].

NF1 is characterized by highly variable inter- and intrafamilial expressivity and multisystemic involvement. Since 1987, clinical diagnosis has been based on clinical-radiological criteria defined by the National Institutes of Health (NIH), which were subsequently reviewed in 2021 [10,11]. The original criteria included: café-au-lait macules (CALMs), seen in more than 95% of infants (Figure 2A,B); axillary and inguinal ephelides (freckling), which appear at around 6–7 years of age (Figure 2A,C); two or more cutaneous neurofibromas, present in about 90% of affected adults (Figure 2D), or one plexiform neurofibroma (pNF; Figure 2E); Lisch nodules (LNs), harmless iris hamartomas visible with a slit lamp in almost all adults but in less than half of children under the age of five; dysplasia of a long or a flat bone, more frequently involving the tibia, with the risk of pseudoarthrosis; the presence of an affected first-degree relative.

The criteria were recently reviewed by a group of international experts (Table 1) [11]. The revision became necessary following the identification of mild NF1 phenotypes associated with specific genotypes and, in 2007, of LGSS, an NF1-like condition with CALMs and freckling, indistinguishable from classic NF1 and caused by mutations in *SPRED1* [12]. LGSS patients do not have any of the NF1-related oncological risks. Given the incomplete NF1 penetrance in young patients, the differential diagnosis between NF1 and LGGS represents a challenge for the pediatrician or the clinical geneticist assessing children. In the case of isolated skin manifestations, molecular analysis of *NF1* and *SPRED1* is required to make a proper diagnosis. The revised diagnostic criteria include both to look for pathogenic variants in the two genes and to specify that, in the presence of only CALMs and freckling, even if typical of NF1, the diagnosis must be questioned and a molecular differential diagnosis of LGSS should be performed. The identification of LGSS highlighted a major limitation of the NIH NF1 clinical diagnostic criteria. Additional features of NF1 are therefore considered helpful criteria for improving early diagnosis, given the high frequency and specificity of the condition (Table 2).

## 2. Clinical Features of NF1 in Children and Adolescents

### 2.1. Cutaneous Findings

CALMs are asymptomatic cutaneous macules of rounded or ovoid shape, with clean and homogeneously pigmented edges, well-defined margins, and dimensions ranging from a few millimeters to tens of centimeters. They are the most frequent clinical sign of NF1. CALMs may be already present at birth or become evident in the first months of life, increasing in number and size up to puberty. Subsequently, they tend to undergo depigmentation, making their identification more difficult in adults. NF1 patients typically exhibit skin manifestations and can present a great number of CALMs, which are widespread on any part of the body, except for palms and soles, and rarely occur on the face. The severity of NF1 has no relationship with the number of CALMs. At least six CALMs or more, with a diameter of 0.5 cm and 1.5 cm in pre- and post-pubertal age, respectively, must be present to constitute a diagnostic criterion. Although CALMs may also be found in the general population, several significant CALMs in prepubertal children raise a strong suspicion of NF1, warranting an annual follow-up. CALMs may also be present in other clinical conditions, including LGSS, LEOPARD syndrome, piebaldism, McCune Albright syndrome, and constitutional mismatch repair deficiency (CMMRD) syndrome. Differently from NF1, in McCune Albright syndrome CALMs typically have irregular or jagged margins. Differential diagnosis is necessary and often challenging [14,15].

Freckles (or freckling) are multiple pigmented macules, 1–3 mm in diameter, similar to peaking café-au-lait spots. They are present in about 90% of adults affected by NF1 and generally appear at around 4 years of age in the axillary and/or inguinal regions. They are suggestive of NF1 only when present in these sites. Despite their similarity to freckles present in healthy individuals, freckles in NF1 patients differ in their atypical location, larger size, darker color, and lack of correlation with sun exposure [15,16].

Nevus anemicus (NA) is a congenital hypopigmented skin abnormality. Although friction or heating can cause redness in normal skin, NA lesions will remain pale. Although the correlation between NA and NF1 is still unclear, NA seems to have a much higher prevalence in NF1 patients than in the healthy population [15]. Some authors therefore suggest that the presence of NA might be helpful in making an early diagnosis of NF1 in young children [17,18].

Juvenile xanthogranuloma (JXG) is characterized by yellowish-pink dome-shaped granulomatous papules usually located on the upper body. It may appear together with CALMs and disappear spontaneously. Patients with NF1 and JXG were previously reported as being at risk of developing juvenile myelomonocytic leukemia (JMML). However, this association has now been confuted, as recently reported by the ERN GENTURIS tumor surveillance guidelines for individuals with NF1 [19,20].

### 2.2. Eye Findings

Eye manifestations may occur in early childhood as optic pathway gliomas (OPGs) or later in childhood and in adolescence as LNs. LNs are iris hamartomas, typically multiple and bilateral, occurring in all areas of the iris surface and visible under slit lamp examination. LNs tend to increase in size and number with age. They are typically asymptomatic, do not cause visual impairment, and do not therefore require treatment [21].

In the last few years, choroidal anomalies (CAs) were described as a new ocular sign characterizing NF1 and were included in the diagnostic criteria [13,22,23,24]. CAs are ovoid bodies composed of proliferative Schwann cells, melanocytes derived from the neural crest, and ganglionic cells around the axons arranged in lamellar patterns, undetectable with conventional ophthalmoscopic examination or in fluorescein angiograms [25]. However, Yasunari et al. found that CAs could be detected by infrared light examination with scanning laser ophthalmoscope in 100% of their NF1 patients [26]. The detection of CAs as a clinical sign of NF1 seems to be also more accurate than the presence of LNs [23,25].

### 2.3. Tumors

Patients with NF1 have an increased risk of developing tumors that primarily involve the nervous system and generally show a benign histology.

OPGs are the most frequent tumors of the central nervous system (CNS) in children affected by NF1 under the age of seven, with an incidence of 15–20% [27]. In 50–70% of cases of NF1, patients are asymptomatic and screening is not recommended. If visual abnormalities such as decreased visual acuity, nystagmus, pupillary pallor, strabismus, campimetric deficit, or proptosis are observed, brain magnetic resonance imagining (MRI) should be performed (Figure 3A,B) [28]. However, the usefulness of brain MRI screening is debated because of the need for sedation in young children. While children older than 7 years can often comply with instructions to remain still, younger children (aged 1–6 years) usually require sedation [29]. The majority of OPGs are grade I pilocytic astrocytomas (World Health Organization classification) and can involve any part of the optic pathway. Due to their occasional spontaneous regression, their course in NF1 can be more indolent than sporadic OPGs [30]. Based on OPG location, a worse visual prognosis is associated with the presence of optic atrophy, multiple visual signs, and loss of visual acuity [30]. A recent meta-analysis reported that visual outcome is even worse if patients undergo surgery or radiotherapy [31]. In untreated OPGs, endocrinological complications are closely related to hypothalamic involvement, making a thorough auxological follow-up of children with NF1 with or without OPG essential [32]. Based on frequency, the most common endocrinological disorders associated with OPGs in NF1 are precocious puberty, excess/deficiency of growth hormone, and diencephalic syndrome [33,34,35].

Low-grade extra-optic tumors commonly involve the brain stem (ependymomas, medulloblastomas, and dysplastic neuroepithelial tumors), and are often diagnosed incidentally (Figure 3C,D). The prognosis associated with these tumors is generally good. NF1 patients with more than one low-grade extra-optic glioma have faster disease progression with a shorter 5-year survival rate than those with only one lesion, regardless of age onset and presence of symptoms [32,36,37].

Spinal neoplasia associated with NF1 can be extramedullary, such as neurofibromas and malignant peripheral nerve sheath tumors (MPNSTs), and intramedullary, including astrocytomas, ependymomas, and gangliogliomas [38].

More than 30% of NF1 patients have pNFs, benign tumors originating from the myelin sheath of the peripheral nervous system and with a preferential location in the trunk (44%), limbs (38%), and head/neck (18%) [5]. The involvement of deep nerves affects the emergence of spinal roots and, in this case, pNFs involve paravertebral spaces (Figure 3E,G,H). pNFs are thought to be congenital and generally asymptomatic lesions [39]. In some patients, however, they can cause serious disfigurement with esthetic problems and neuropathic symptoms, reach a considerable size, produce excessive growth or erosion of adjacent tissues, and functional disorders depending on the body district affected (breathing difficulties or visual disturbances) (Figure 1E) [40]. Until recently, the only therapy available was subtotal surgical excision, which has a high frequency of regrowth and carries the risk of iatrogenic neurological conditions and intraoperative hemorrhage [41]. Recently, the selective enzyme mitogen-activated protein kinase inhibitor selumetinib was found to be effective in reducing the size of pNFs, improving quality of life by controlling pain and associated neurological deficits [42]. In the United States, selumetinib (trade name Koselugo) has been approved by the US Food and Drug Administration for the treatment of children of two or more years of age with inoperable and symptomatic pNFs. In Europe, it has been designated as an orphan drug (EMEA/H/C/005244) under conditional marketing authorization by the European Medicines Agency.

MPNSTs are soft tissue sarcomas. NF1 patients have an approximately 10% lifetime risk of developing an MPNST, which tends, although not exclusively, to occur in a pre-existing pNF [5,43]. In a large cohort of NF1 patients, King et al. reported a higher than expected relative risk of developing MPNSTs [44]. A cross-sectional study showed that 1–2% of NF1 patients develop MPNSTs [45]. According to King et al., the average 10-year annual incidence of MPNSTs between the second and fifth decade of life was found to range between 0.0013 and 0.0068 MPNSTs per patient year [44]. In NF1, MPNSTs tend to occur earlier (at 20–40 years) and seem to have a worse 5-year survival rate from diagnosis compared to sporadic cases [43,46,47,48]. They are preferentially located in the lower limbs and pelvis, involving the sciatic nerve [47]. MPNSTs are resistant to radiotherapy and chemotherapy and have a poor prognosis and a high mortality rate, representing the leading cause of death in patients with NF1 [43]. The radical surgical excision of non-metastatic MPNSTs is the only potentially curative therapy currently available.

### 2.4. Other Complications Affecting the Nervous System

Neurofibromatosis bright objects, presently named as focal areas of signal intensity, are benign brain lesions appearing hyperintense on unilateral or bilateral T2-weighted images with blurred edges, not detected by contrast medium (Figure 3A). They are characteristic but not pathognomonic of NF1 and can be found in the cerebellum, brain stem, thalamus, and basal ganglia in 70% of children with NF1. They appear at around 3 years of age and tend to disappear between the ages of 20 and 30 [49,50]. Lesions with similar radiological characteristics and benign evolution can also be found in the spinal cord [51] (Figure 4A,B).

The majority of NF1 patients are of normal intelligence, although it may be lower than that of both unaffected siblings and the general population. Up to 80% of affected children have learning difficulties or behavioral problems; characteristics of autism spectrum disorder are found in 30% of patients [52,53,54]. Intellectual disability is observed in 6–7%, twice the frequency of the general population [55]. Deficits in visual–spatial performance, social competence, and attention are frequent and are often associated with problems of motor, executive, memory, and language function [56].

Epilepsy has an overall incidence of around 5.4%, which seems to be slightly lower in pediatric age (3.7%) [57]. At early age, it is mostly secondary to brain tumors, hydrocephalus (Figure 4C,D), and Moyamoya vasculopathy (Figure 4D,E) as well as cortical malformations and mesial temporal sclerosis. A literature review suggested that less than one third of NF1 patients have drug-resistant epilepsy, in line with the general population [57]. For NF1-related and lesional forms of epilepsy, surgical treatment is recommended [58].

Hydrocephalus in patients with NF1 is usually obstructive and secondary to tumors (OPGs or other tumors of the CNS) (Figure 4C,D) [59,60]. The pattern depends on the exact location of the obstruction, leading to bi- and triventricular forms. Non-tumor etiologies include hamartomas, via involvement of the cerebral aqueduct system and synechiae of the superior medullary velum [60]. Third ventriculocisternostomy is a safe treatment for selected NF1 patients with obstructive hydrocephalus. In other cases, shunts and stents are necessary [60,61].

Chiari I malformation is characterized by the herniation of the cerebellar tonsils through the foramen magnum. Few cases are reported in the literature in association with NF1, and in more than 8% of these patients, it was asymptomatic. The most frequent symptom is recurrent periodic headache (69%), syringomyelia (30%), sensitivity disorders, and hypoesthesia [62].

### 2.5. Vascular Diseases

Neurofibromin is expressed in the vascular endothelium and in vascular smooth muscle. The loss of neurofibromin by some means underlies NF1 vasculopathy, which can affect any blood vessel and district. In the extracranial circulation, stenosis mostly involves the renal artery [63]. Hypertension is relatively common in NF1, with an estimated frequency of 16–19%, depending on literature source [13]. It is generally “essential”, can develop at any age, and increases with age. However, it can also be present as the result of renal artery stenosis, aortic coarctation, or other vascular lesions, such as mid-aortic syndrome, and can be caused by neuroendocrine tumors (pheochromocytoma, paraganglioma, and neuroblastoma) [64,65,66,67].

Moyamoya vasculopathy is the most frequent cerebral arterial disease in NF1 (3–5%) and is characterized by a progressive occlusion of cerebral vessels (terminal branches of the internal carotid and arteries of the circle of Willis) [65]. In Japanese, the name evokes the characteristic “puff of smoke” angiographic appearance resulting from the formation of an anomalous collateral network of small vessels [68]. Brain MRI and angio-MRI (ARM) are useful and non-invasive methods for diagnosing and monitoring the disease (Figure 4E,F). Although cerebral angiography is still the gold standard for diagnosis of Moyamoya vasculopathy, it is not essential when MRI and ARM show the typical findings of the disease. Perfusion MRI performed using the dynamic susceptibility contrast technique together with cerebral perfusion tomoscintigraphy with technetium (SPECT-99mTc-HMPAO) allows for non-invasive integration of previous techniques, documenting cerebral hemodynamics [69]. Although at least half of the forms of Moyamoya vasculopathy are asymptomatic, onset in children is characterized by ischemic events (transient ischemic attacks and ischemic heart attacks), focal epilepsy, or headaches [70]. The risk of progression and neurological damage is an indication for revascularization surgery. An indirect technique using multiple burr holes is as effective as the traditional bypass technique with superficial anastomosis of the temporal artery with the middle cerebral artery, and has the advantage of intervening bilaterally at same time [71]. The pathogenesis of Moyamoya vasculopathy is not yet completely elucidated, and in NF1, certainly independent of the type of germline mutations in *NF1* [72]. However, modifier genes such as *MRVI1* and *RNF213* were recently suggested to have a potential synergistic effect [73,74,75].

### 2.6. Cardiovascular Involvement

In NF1, the prevalence of congenital heart disease is highly variable, ranging from 0.4% to 6.4% [76]. Pulmonary valve stenosis (PVS) is the most frequent disorder, followed by mitral valve anomalies. Adults with NF1 can also develop pulmonary hypertension, often in association with late-onset pulmonary parenchymal disease causing potentially serious complications [77]. Intracardiac neurofibromas are very rare [78].

Arterial hypertension affects approximately 15% of the pediatric population with NF1 [13,64,79,80,81]. Excluding the correlation with renal artery stenosis, coarctation of the aorta and mid-aortic syndrome, which figure pediatric forms of arterial hypertension, hypertension is usually essential in NF1 [82]. Among these secondary forms, hypertension can also be due to secreting paraganglioma/pheochromocytoma, which are extremely rare in pediatric age [83].

### 2.7. Skeletal Manifestations

Several skeletal manifestations are associated with NF1 and include: short stature (for which specific percentile curves have been produced) [84,85], macrocephaly, joint laxity, dysplasia of the long bones (tibia and fibula), chest deformities typically represented by sternum excavatum or carinatum (Figure 4G,H), dysplasia of sphenoidal wings or other flat bones, vertebral dysplasia, osteofibromas, and spinal rotoscoliosis [84,86,87,88].

In addition to idiopathic scoliosis, which has a similar clinical presentation and natural history to sporadic forms, dystrophic scoliosis can be observed in NF1 and presents with early onset (6–8 years), rapidly progressive evolution, and involvement of a few vertebrae with severe curvature [89]. Vertebral scalloping and dura mater ectasia are usually observed in the dysplastic form (Figure 3E,F). Co-occurrence of paravertebral neurofibromas in the scoliotic tract is common in dysplastic forms and often requires surgical correction with spinal fusion [87].

Generalized osteopenia and early osteoporosis with an increased risk of fracture are common in children affected by NF1 and are attributable to abnormal bone metabolism, altered remodeling processes, and, partly, to a tendency to hypovitaminosis D [87,90,91].

### 2.8. Miscellaneous

Children with NF1 may have oral manifestations due to the presence of pNFs in the trigeminal area or due to a greater risk of caries [92,93]. Other characteristics associated with NF1 are Noonan-like signs such as facial dysmorphism, short stature, ligamentous hyperlaxity, soft palms of the hands and soles of the feet [85,94]. Pubertal delay is common [95].

## 3. Follow-Up and Management

The medical follow-up of pediatric patients with NF1 is based on an active collaboration between multiple healthcare professionals using a multidisciplinary approach. Periodic monitoring is recommended as soon as a diagnosis of NF1 is suspected. For patients with a high-risk of more severe complications, annual or semi-annual evaluation is recommended, while those with a mild phenotype and low risk of complications may be evaluated every two to three years. In the literature, mild phenotype refers to those patients without oncological features, potentially life-threatening disorders, or conditions associated with highly morbid complications of NF1 (e.g., OPG, pNFs, in some cases disfiguring, and surgical scoliosis). Severe phenotype refers to patients with the latter features or with intellectual impairment, brain tumors, or high-burden age-related cutaneous neurofibroma. One of the first severity scales for NF1 was published by Riccardi and Kleiner in 1977 [96]. However, no scales have yet been approved for clinical stratification of NF1 patients, useful in determining the risk of an individual patient or in designing a more personalized follow-up schedule of tests and visits. Today, any such scale should include screening for genetic germline pathogenetic variants in *NF1*. Annual clinical follow-up allows for the early diagnosis of complications, reduction in morbidity, and improvement in quality of life. A complete physical examination, including blood pressure measurement, should be carried out during each visit. Routine invasive exams are not recommended, but should only be undertaken after extensive clinical assessment during a specific follow-up visit. Table 3 summarizes the screening modalities to be performed in the follow-up of patients with NF1, as suggested by the NF France Network in 2020 after an extensive literature review [13].

## 4. Genetics

### 4.1. Genotype–Phenotype Correlations

The manifestations of NF1 are age dependent, extremely variable, and partly overlapping with other RASopathies, such as Noonan syndrome and LGSS. The vast majority (90–95%) of *NF1* causative variants are intragenic, while less than 10% are deletions involving the entire gene and flanking genomic regions [82,97,98,99]. To date, five genotype–phenotype correlations have been identified and are summarized in Figure 5 and Figure 6.

The first identified genotype–phenotype correlation was 17q11.2 microdeletion syndrome (MIM 613675), involving *NF1* and adjacent genes. The architecture of this region, with flanking repeated sequence elements (segmental duplications), favors deletion events in the germline of a healthy parent or during mitosis [97].

Different types of microdeletions (type 1, 2, 3, and atypical) are classified by size, location of breakpoints, and deleted genes [100]. The most frequent (70–80% of cases) is a type 1 microdeletion (1.4 Mb) that includes 14 protein-coding genes and four microRNA-coding genes. It usually occurs de novo, as the result of a germline rearrangement. About 10% of microdeletions are type 2 (1.2 Mb) and involve 13 protein-coding genes. Breakpoints involve *SUZ12* and its *SUZ12P1* pseudogene, which are often of postzygotic origin and are therefore associated with somatic mosaicism [101]. Type 3 microdeletions (1.0 Mb) occur in 1–4% of cases and involve nine protein-coding genes [102]. Atypical deletions with non-recurrent breaking points are rare and may have a germinal or postzygotic origin [97,103,104].

Compared to NF1 patients with intragenic mutations, patients with the 17q11.2 microdeletion typically have a higher number of cutaneous and subcutaneous neurofibromas (more than 1000) and earlier onset (prepubertal). They can also present multiple spinal neurofibromas (mainly plexiform) and more frequently show intellectual disability [97]. They also have a higher risk of developing MPNSTs and often have distinctive facial features: a wide neck, hypertelorism, epicanthal folds and down-slanting palpebral fissures, broad and rough nose, and coarse facial features [97]. About half of patients with a type 1 *NF1* microdeletion have overgrowth in childhood and a tall stature in adulthood (≥94th percentile) [97,103]. Heart defects are significantly more common; however, the type and frequency of observed cardiovascular defects are quite heterogeneous [78]. Although there is no direct correlation between epilepsy and *NF1* mutations, it appears to be more prevalent in patients with the 17q11.2 microdeletion, especially types 1 and 3, possibly associated with deletion of *MIR193A* [105].

The other observed genotype–phenotype correlations involve single amino acid substitutions or in-frame deletions (Figure 5 and Figure 6). The in-frame deletion of three nucleotides in exon 17 of *NF1* (c.2970_2972delAAT) removes the methionine at position 992 (p.Met992del) and is associated with a milder NF1 phenotype, characterized by CALMs and freckling with the absence of skin neurofibromas and subcutaneous or superficial pNFs [106,107]. A recent study of 135 NF1 patients with this mutation confirmed a lower frequency of LNs and the absence of spinal neurofibromas and symptomatic OPGs. However, 4.8% of the NF1 patients had low-grade, asymptomatic extra-optic brain tumors, and 38.8% had an intellectual disability and/or cognitive impairment [106].

A phenotype characterized by CALMs, freckling, the absence of histologically proven neurofibromas and OPGs, and a lower frequency of LNs was associated with the substitution of arginine 1809 (p.Arg1809) [108,109,110]. Our group also very recently observed the absence of choroidal abnormalities in our NF1 patients with the Arg1809 substitution [111]. Such patients have a higher prevalence of short stature, learning disabilities, PVS, and Noonan-like characteristics [11,108,109,110].

Both these genotype–phenotype correlations are potentially clinically indistinguishable from LGSS, an NF1-like autosomal dominant disorder secondary to heterozygous mutations in *SPRED1*, which is also a negative regulator of the RAS-MAPK pathway [12,98,112]. To date, LGSS has never been associated with the oncological complications found in NF1 [11].

Missense variants in codons 844–848 of *NF1* are associated with a higher prevalence of symptomatic spinal neurofibromas (p.Leu844, p.Cys845, p.Ala846, p.Leu847, p.Gly848) and superficial pNFs (p.Cys845, p.Ala846), and an increased risk of developing MPNSTs and other malignancies including rhabdomyosarcomas, JMML, and neuroblastomas caused by the p.Leu847Pro variant. OPGs (p.Leu844) and skeletal abnormalities leading to significant morbidity are also common in this cohort of patients [113].

Missense variants in p.Met1149, p.Arg1276, and p.Lys1423 were recently associated with an NF1 phenotype frequently characterized by Noonan-like features [114]. NF1 patients carrying substitutions in p.Met1149 presented a milder phenotype with a predominance of skin manifestations, thoracic abnormalities, short stature, macrocephaly, and intellectual disability, with the absence of symptomatic spinal pNFs and OPGs. In contrast, variants in p.Arg1276 and p.Lys1423 are associated with a more severe phenotype and a higher prevalence of cardiovascular anomalies, including PVS. Amino acid changes in p.Lys1423 are associated with visible pNFs, LNs, and skeletal abnormalities, while symptomatic spinal tumors occur less frequently and OPGs are rare. Substitutions affecting p.Arg1276 were found to be the second leading cause of symptomatic spinal neurofibromas after those involving p.Gly848, and are associated with cardiac anomalies, including PVS, skeletal anomalies, the absence of OPGs, and poor development of cutaneous neurofibromas [114]. Patients with p.Gly848 substitutions were found to present fewer pigmentary manifestations and 70% of patients over 9 years old had symptomatic spinal neurofibromas [113,114].

NF1 patients with in-frame mutations, which do not alter the reading frame, have a six times greater risk of developing PVS than those with out-of-frame mutations [115]. De Luca et al. hypothesized that mutations in the regulatory portions of *NF1* may have a pathogenetic role in NF1-Noonan syndrome (NFNS; MIM 601321) [116]. NFNS and Watson syndrome (MIM 193520) exhibit a higher prevalence of non-truncating variants in *NF1* (exons 24 and 25) and are two phenotypic subtypes of NF1 [115,116]. NFNS is characterized by a low incidence of pNFs, skeletal abnormalities, tumors, hypertelorism, ptosis, low-set ears, and congenital heart defects [116], while Watson syndrome presents with PVS, CALMs, and intellectual disabilities [117].

Overall, the genotype–phenotype correlations identified to date only partially explain the clinical variability of NF1, which could also be determined by trans-acting events such as alternative splicing or the effect of modifier genes [74,118]. The possibility of predicting the phenotypical features of the disease, even in a small proportion of cases, makes molecular diagnosis a valuable tool in designing the best follow-up regimen for patients with NF1.

### 4.2. Association with Other Genetic Disorders

Likely due to its high prevalence, NF1 has been associated with other genetic conditions that can complicate the phenotype of patients, although the main features of NF1 remain well recognizable and preserved. NF1 association with Noonan syndrome (*PTPN11*) [119,120], Huntington’s disease (*HTT*) [121], congenital myopathy (*RYR1*) [122], hereditary breast cancer (*BRCA1*) [123], multiple endocrine neoplasia type 2 (*RET*) [124], and Jalili syndrome (association between conical rod-dystrophy and amelogenesis imperfecta) are all described [125]. Down syndrome and NF1 were also observed together, as well as a pathogenic de novo variant of *NF1* and de novo deletion of chromosome 20q11.23 [126,127]. Further cases of NF1 and other concurrent monogenic disorders associated with causative variants in *RAB39* or *MEIS2* were also reported by our group [128,129].

### 4.3. Molecular Diagnosis

The identification of mutations in *NF1* is complicated by the large size of the gene, the absence of mutational hotspots, and the presence of pseudogenes. Mutations mainly involve single base substitutions or insertions/deletions of one or more nucleotides, and less frequently by deletions/duplications of one or more exons and microdeletions of the entire *NF1* gene [98]. Point mutations are now being investigated using next generation sequencing (NGS) techniques, which are much more sensitive and less expensive than Sanger sequencing, which is still used to validate NGS results [98]. NGS is also able to provide a molecular differential diagnosis with other RASopathies [98,130], while the screening of deletions/duplications and microdeletions can be performed by multiplex ligation-dependent probe amplification (MLPA) [98,131]. RNA analysis can be useful to assess NGS- and MLPA-negative patients, due to the possible presence of deep intronic mutations with an effect on splicing, as well as to better understand the effect of genomic variants found in the affected subjects, potentially leading to the definition of new genotype–phenotype correlations [98,132].

## 5. Conclusions

Continuous improvements in our clinical and molecular knowledge of NF1 has led to a better understanding that not all variants have the same effects. The growing number of genotype–phenotype associations is slowly but profoundly changing the clinical and genetic approach to NF1 patients. As genotype–phenotype correlations continue to increase, genotype-driven precision medicine could mark a major turning point in the management of NF1 by improving disease surveillance and patient stratification, possibly opening the way to novel therapeutic approaches.

## Figures and Tables

**Figure 1 cancers-15-01217-f001:**
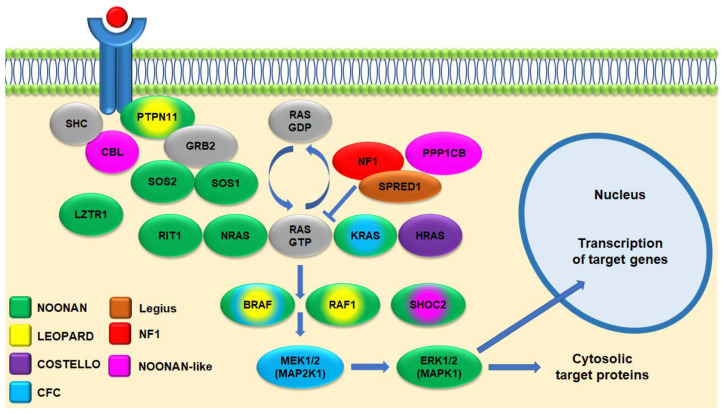
Schematic representation of the RAS/MAPK pathway in RASopathies. Germinal mutations in genes encoding different components of the pathway can cause overlapping phenotypes, color coded according to the legend.

**Figure 2 cancers-15-01217-f002:**
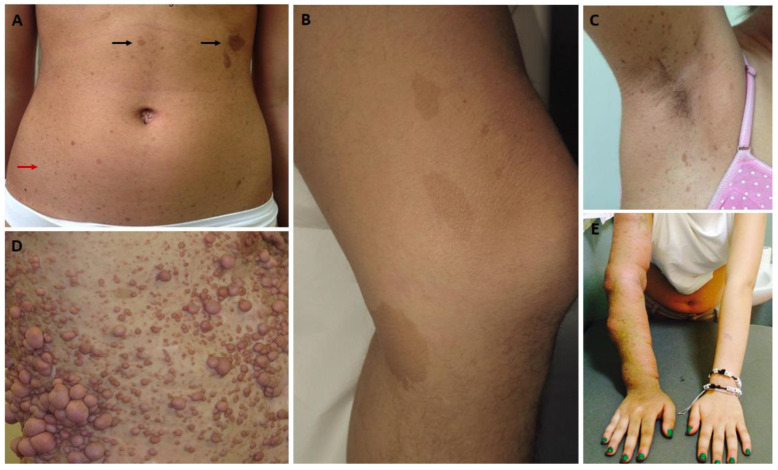
Typical manifestations of neurofibromatosis type 1 (NF1): café au lait macules characterized by linear margins (**A**,**B**), homogeneous color, and rounded/oval shape ((**A**); black arrow), with dimension > 1.5 cm in post-pubertal age; atypical freckles on the abdomen ((**A**); red arrow); typical freckles in the axillary fold (**C**); multiple cutaneous neurofibromas typically found in adulthood—note the pinkish color and rounded shape (**D**); extensive plexiform neurofibroma involving the entire upper limb, with hyperchromic skin lesions in the affected area and secondary deformity (**E**).

**Figure 3 cancers-15-01217-f003:**
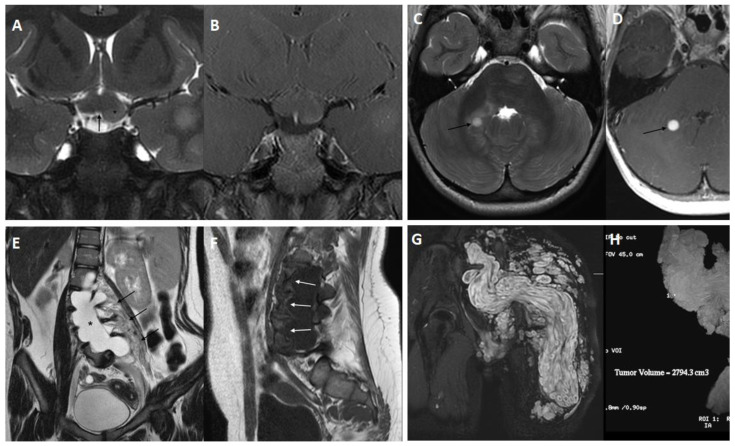
Coronal T2-weighted (**A**) and T1-weighted MR images after fat suppression contrast (**B**) showing a thickened optic chiasm (black arrow) with a lobulated mass-effect lesion (asterisk) on the left showing a hyperintense signal and a linear contrast enhancement. T2 axial TSE sequences (**C**,**D**) showing a hyperintense rounded focal lesion (black arrow) with minimal marked and homogeneous vasogenic edema (black arrow) in the right middle cerebellar pedicle in line with a likely low-grade lesion. Coronal T2-weighted (**E**) and sagittal T1-weighted (**F**) sequences of lumbosacral spine showing a plexiform neurofibroma (black arrows), marked dural ectasia (asterisk) associated with significant scalloping of the posterior lower lumbar vertebral bodies (**white arrows**). Coronal STIR images with volumetric reconstruction (**G**,**H**) showing a voluminous plexiform neurofibroma arising from the lumbo-sacral plexus and extending into the left thigh region.

**Figure 4 cancers-15-01217-f004:**
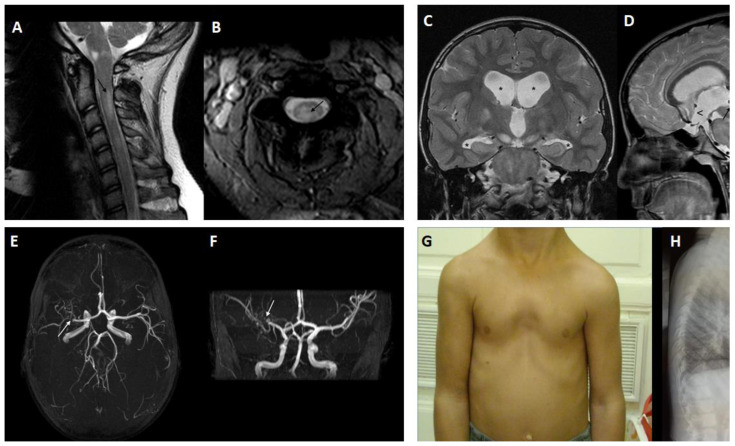
Sagittal (**A**) and axial bFFE STIR-FSE (**B**) MR images of the cervical spine showing at C2-C3 level a focal area of hyperintensity located on the right posterior side of the medulla, compatible with UBO. T2-weighted coronal sequence showing dilation of the frontal and temporal horns of the lateral ventricles (asterisk) (**C**). T2-weighted sagittal MR image (**D**) showing aqueduct stenosis (black arrow) with widening of the hypothalamic recess. Angio-MR image in a boy with Moyamoya vasculopathy (**E**,**F**): the 3D time-of-flight technique with sagittal and coronal maximum intensity projection reconstructions shows focal stenosis of the M1 segment of the right middle cerebral artery (white arrow); “Moyamoya-type” collateral vessels are visible. Sternum excavatum (**G**) and profile X-ray showing malformation of the sternum (**H**).

**Figure 5 cancers-15-01217-f005:**
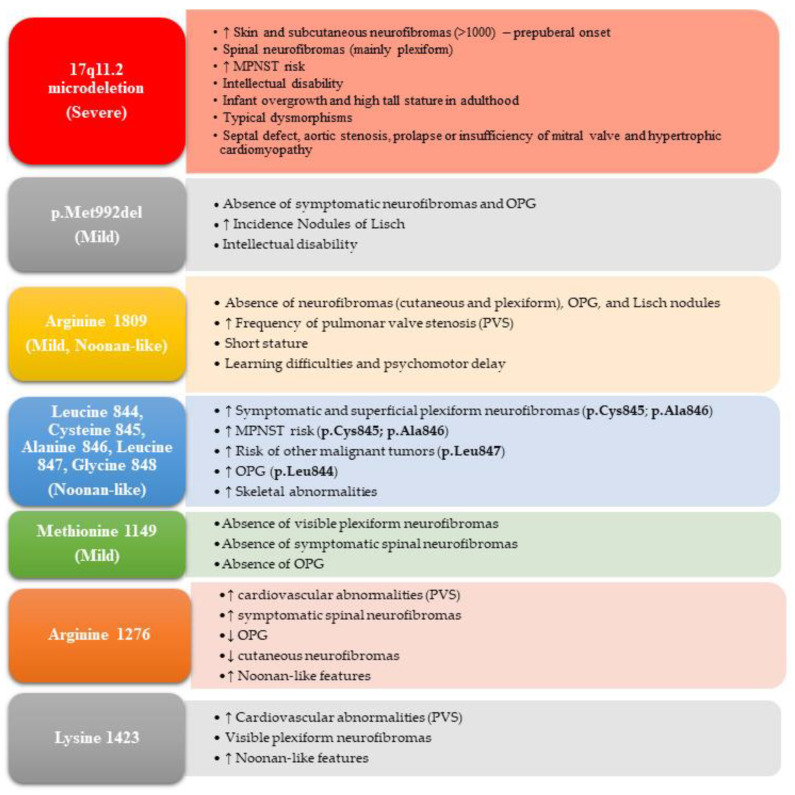
Genotype–phenotype correlations known to date for *NF1*. Major phenotypic characteristics significantly more (↑) or less (↓) present, or possibly absent, are listed. Amino acid changes at specific positions are grouped according to the observed phenotypic features.

**Figure 6 cancers-15-01217-f006:**
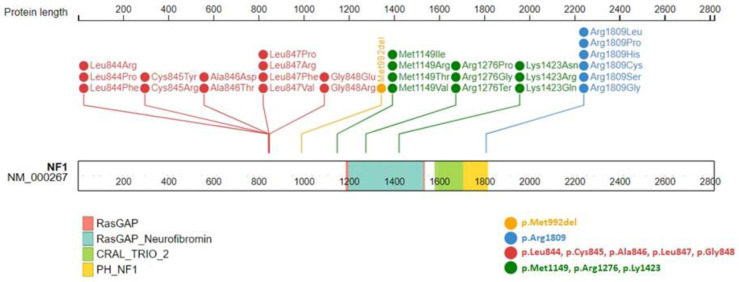
Graphical distribution of *NF1* missense variants currently associated with different classes of mild to severe phenotypes: deletion of methionine 992 (orange); six different substitutions of arginine 1809 (blue); 14 substitutions involving leucine 844, cysteine 845, alanine 846, and glycine 848 (red); 13 substitutions of methionine 1149, arginine 1276, and lysine 1423 (green). All these variants were annotated in Clinvar and refer to the NM_000267.3 isoform.

**Table 1 cancers-15-01217-t001:** Revised diagnostic criteria for neurofibromatosis type 1 as reported by Legius et al. [11].

Revised Diagnostic Criteria for Neurofibromatosis Type 1 (NF1)
A: The diagnostic criteria for NF1 are met in an individual who does not have a parent diagnosed with NF1 if two or more of the following are present:
• Six or more café-au-lait macules over 5 mm in greatest diameter in prepubertal individuals and over 15 mm in greatest diameter in post-pubertal individuals ^1^
• Freckling in the axillary or inguinal region ^2^
• Two or more neurofibromas of any type or one plexiform neurofibroma
• Optic pathway glioma (OPG)
• Two or more iris Lisch nodules identified by slit lamp examination or two or more choroidal abnormalities (CAs)—defined as bright, patchy nodules imaged by optical coherence tomography (OCT)/near-infrared reflectance (NIR) imaging
• A distinctive osseous lesion such as sphenoid dysplasia ^3^, anterolateral bowing of the tibia ^4^, or pseudarthrosis of a long bone ^5^
• A heterozygous pathogenic NF1 variant with a variant allele fraction of 50% in apparently normal tissue such as white blood cells
**B: A child of a parent who meets the diagnostic criteria specified in A merits a diagnosis of NF1 if one or more of the criteria in A are present**
^1^ If only café-au-lait macules and freckling are present, the diagnosis is most likely NF1 but exceptionally the person might have another diagnosis such as Legius syndrome. At least one of the two pigmentary findings (café-au-lait macules or freckling) should be bilateral. ^2^ If only café-au-lait spots and freckles are present, the diagnosis is probably NF1 but exceptionally the person may have another diagnosis such as Legius syndrome. At least one of the two pigment findings (café-au-lait spots or freckles) should be bilateral. ^3^ Sphenoid wing dysplasia is not a separate criterion in case of an ipsilateral orbital plexiform neurofibroma. ^4^ Congenital bowing is usually associated with thickening of the cortex of the long bone. ^5^ Pseudarthrosis is usually preceded by congenital curvature of a long bone and only rarely by thinning of the cortex.

**Table 2 cancers-15-01217-t002:** Typical additional characteristics of NF1 according to Bergqvist et al. [13].

NF1 Additional Features	
Dystrophic scoliosis	Typical and should be clearly defined
Juvenile xanthogranuloma	Present in up to 30% of very young patients Not reported in Legius syndrome
Anemic nevus	Present in up to 50% of children with NF1 and may allow for clinical diagnosis at young age, but may also be seen in other conditions and in the general population
Choroidal anomalies	Present in 60–70% of children with NF1
Focal areas of signal intensity (FASI)	Previously known as unidentified bright objects (UBOs) Common neuroradiological findings in brain MRI and may disappear over time

**Table 3 cancers-15-01217-t003:** Summary of screening procedures to be performed in the medical follow-up of patients with NF1, as recommended by the NF France network and reported by Bergqvist et al. [13].

Screening for Major NF1 Complications			
Sought Complications		Affected Patients	Screening Modality
Dermatological manifestations	Subcutaneous, internal, and plexiform NF: malignant transformation? Esthetic or functional problems?	Children, adults	Clinical examination: Pain, neurological deficit, increase in size, functional and psychological repercussions. Additional examinations: optional Indications: suspicion of malignancy, preoperative, internal NF risk factor
	Juvenile xanthogranuloma (JXG)	Children	Physical examination: If JXG present: palpation of ganglionic areas and complete blood count *
Orthopedic manifestations	Bone dysplasia and pseudarthrosis of the long bones, fractures	Children, adults	Clinical examination: search for gibbosity, bone deformity. X-ray if abnormalities found on clinical examination.
	Scoliosis	Children, adults	Physical examination Additional examinations (optional): Front and profile X-ray views of the spine if clinical abnormalities found (1st line) MRI should be reserved for forms with vertebral and/or costal dysplasia (expert consensus) Pulmonary function tests to evaluate the impact of severe scoliosis
	Bone mineralization disorder, osteoporosis	Children, adults	Consider bone densitometry scans based on clinical examination, vitamin D levels and X-ray results
Endocrinological manifestations	Pubertal and growth disorders	Children	Follow pubertal development and the growth curve, measure head circumference
Cardiac and vascular manifestations	Essential and secondary hypertension	Children, adults	Physical examination: Blood pressure measurement at each consultation (at least annually), discuss the possibility of ambulatory measurement. Look for signs suggestive of pheochromocytoma. Additional examinations if high blood pressure. As a first-line examination: angio-CT scan of the renal arteries and abdominal CT. Plasma and/or urinary determination of metanephrines in adults.
	Cardiac abnormalities	Children, adults	Clinical examination
	Hemorrhagic manifestations	Children, adults	Assess hemostasis before any surgical, dental, or obstetric procedure.
Pain, psychological repercussions, quality of life		Children, adults	Clinical examination Offer psychological counseling, pain specialist referral
Otolaryngologic manifestations	Deafness, neurinoma, phonatory disorder, laryngeal NF	Children, adults	Otolaryngologic examination with tuning fork
Neurological manifestation	OPG	Children	Interview: repeated falls leading to suspicion of decrease visual acuity or visual field amputation Neurological and ocular examination: strabismus, nystagmus, low visual acuity, neurological deficit, signs of intracranial hypertension. Early puberty, deviation from the growth curve, measurement of head circumference Ophthalmological screening at least once per year until the age of 13 years and then if signs appear. MRI of the optic and cerebral pathways is not systematic and should be performed only if suspicion of OPG
	Epilepsy, hydrocephalus, intracranial, hypertension, stroke, headache	Children, adults	Neurological examination Cerebral MRI and electroencephalogram guided by the abnormalities detected on clinical examination
	Developmental delay, learning difficulties, behavioral problems	Children	Evaluation of psychomotor development and academic proficiency at each consultation Search for learning difficulties Comprehensive neuropsychomotor assessment before entering elementary school, support for school integration.
	Medullary and nerve compression, peripheral neuropathy, socio-professional integration	Adults	Clinical examination
Cancers	MPNST (60% of cancers in NF1 patients)	Children, adults	Clinical examination: recent increase in size of plexiform NF, appearance of pain. Additional examinations if signs appear. If high NF1 score: screening for internal neurofibromas by imaging (preferably by MRI).
	All other cancers	Children, adults	Clinical examination: asthenia, high blood pressure, intracranial hypertension symptoms, abdominal mass, bladder signs, appearance of mass, compressive syndrome Screening identical to that of the general population except for earlier breast screening (>40 years)

* Note that is not recommended by the recent GENTURIS guidelines [19].

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
