# Peer review of "Neurofibromatosis Type 1: Pediatric Aspects and Review of Genotype–Phenotype Correlations"

_cancers, 2023, doi:10.3390/cancers15041217_

Round 1

Reviewer 1 Report

The authors have presented a review of the pediatric aspects of NMF1 and a review of genotype-phenotype correlations. Overall this is a reasonable effort but the authors must be clearer on their epidemiology and quote source references not other reviews. The English does need some improvement as pointed out in several places below

Specific points

1.       Title: Neurofibromatosis type 1: pediatric(s) aspects and review of genotype-phenotype correlations’ lose the ‘s’ in pediatric

2.        Simple summary: ‘In the last few years, an increasing number of genotype-phenotype correlations has(ve) been described for the Neurofibromatosis type 1..’ ‘has’ should be ‘have’

3.       ‘: The Neurofibromatosis type 1 (NF1) is an autosomal dominant condition with a birth in- cidence of approximately 1:2000-3000, caused by (mutations) in NF1,..’ -The preferred term is now ‘germline pathogenic variants’

4.       ‘In this scenario of extreme phenotypic variability, there are some geno-type-phenotype associations that we need to recognize because (strongly impacting) on genetic counseling and prognostication of the disease.’ - (strongly impacting)-rephrase to ‘they strongly impact’

5.       ‘The incidence is 1/2000-3000 individuals in the general population [1-3]’ -You must define ‘incidence’. Here you mean birth incidence. Please also cite source references rather than reviews so references 1 and 2 should be replaced

6.       ‘(In) About 50% of cases, it is caused by de novo mutations of NF1, on chromosome 17q11.2, encoding neurofibromin,..’ -English

7.       ‘NF1 is characterized by highly variable inter- and intra-familiar expressivity and multi-systemic involvement. Clinical diagnosis has always been based on the clinical-radiological criteria defined in 1987 by the National Institutes of Health [8].’ -NOT always. How was it made before 1987? There are now new revised criteria that should be quoted so the authors should say since 1987 until 2021 and cite new reference. Although you do quote these later the use of ‘always’ here is inappropriate

8.       ‘. CALMs may also be present in other clinical conditions, including LGSS, LEOPARD syndrome, piebaldism, McCune Albright syndrome, and Mismatch Repair Deficiency (MMR-D) syndrome’ The latter is usually referred to as Congenital’ in both that and McCune Albright syndrome the CALMS have a usually irregular border unlike NF1

9.       ‘Nevus anemicus (NA) is a congenital hypopigmentated skin abnormality’ -why is this anemic nevus in the table- be consistent

10.   ‘In the last few years, a new ocular sign represented by choroidal anomalies (CAs) has been characterized in NF1 and included in the diagnostic finding [1].’ -is ref 1 really the most appropriate here?

11.   ‘OPGs are the most frequent tumors of the central nervous system (CNS) in children under the age of 7, with an incidence of 15-20% [19]’ -This is only in children who have been imaged. Over 66% of these are asymptomatic c and never become symptomatic. This figure needs context as MRI scanning of the head is not generally recommended in asymptomatic children with NF1

12.   ‘However, almost half of the patients can develop clinical symptoms or signs (visual acuity deficit, pupillary pallor, strabismus, campimetric deficit, proptosis) [20]’ -I strongly disagree with this statement. It is NOT over half. That would mean >10% of NF1 children develop symptomatic OPG that is simply not true based on large epidemiological studies. https://www.ncbi.nlm.nih.gov/pmc/articles/PMC6849871/ the paper you quote is a biased sample of mostly symptomatic OPG at diagnosis and not reflective of a total population or indeed of a ‘scanned’ population

13.   ‘With a 10% of possibility throughout life of evolving in MPNSTs, pNFs represent a high-risk condition in both morbidity and mortality [2,33]’ -Again the epidemiology is a bit suspect. It is by no means that all MPNST grow from plexiform tumors. The 10% figure is the lifetime risk in the entire population this might mean a higher than 10% risk in someone with a plexiform tumor

14.   ‘MPNSTs are soft tissue sarcomas. In NF1, they have an incidence of about 8-16% and tend to occur earlier (20-40 years) respect to sporadic cases [34-36].’ -You need to define incidence. This is a lifetime risk. Please give an indication of what the childhood risk of MPNST is as that is the subject of the review

15.   Section on vascular ‘Hypertension is relatively common in NF1, with an estimated frequency of 16–19% depending on literature source [1]. It is generally "essential", with development at any age and increasing with age. ‘Moyamoya vasculopathy is instead the most frequent cerebral arterial disease in NF1’ -Please give a figure for childhood as this review is about pediatric aspects. Both hypertension and Moyamoya are mostly adult onset

16.   ‘The pathogenesis of MMS, associated or not to NF1, is unclear, but certainly independent of the type of germline mutations in NF1 [60].’ -Define MMS please. This is a long section on a condition that is rare in childhood but the audience needs to know how rare

17.   ‘Excluding the correlation to renal artery stenosis, it is still debate(able) whether a real connection exist(s),’ If the prevalence is 15% then surely there has to be. Either 15% is wrong-most likely or there is a link as renal artery stenosis is rare

18.   Table ;’ juvenile xanthogranuloma (JXG) Children recs: Physical examination If JXG present: palpation of ganglionic areas and complete blood count’ -This is controversial and NOT recommended by the AACR guidelines or more recent GENTURIS. You need to make this clear

19.   ‘To date, five genotype-phenotype correlations are known and summarized in Figure 5 and 6.’ -The variants should be properly described. What does Met1149 and Arg1276 mean?

20.   ‘Furtherly, we very recently observed CA absence in our NF1 patients with the Arg1809 substitution [91]’ -better to spell out CA here

Author Response

Manuscript revision, round #1

Reply to reviewer #1

We have carefully considered the comments of the reviewer and revised our manuscript according to their suggestions. Our detailed point-by-point reply is reported below and typed in italics.

The authors have presented a review of the pediatric aspects of NF1 and a review of genotype-phenotype correlations. Overall, this is a reasonable effort but the authors must be clearer on their epidemiology and quote source references no other reviews. The English does need some improvement as pointed out in several places below.

We are grateful to the reviewer for their positive comments. We agree that some data need to be clarified and better specified, and that some references need to be changed. We apologize for these issues. The manuscript has now been thoroughly edited for English grammar and style by a professional English-language editor (Catherine J. Fisher). We are confident it is acceptable for publication in the present form.

Specific points

  1. Title: Neurofibromatosis type 1: pediatric(s) aspects and review of genotype-phenotype correlations’ lose the ‘s’ in pediatric

We have deleted the ‘s’.

  1. Simple summary: ‘In the last few years, an increasing number of genotype-phenotype correlations has(ve) been described for the Neurofibromatosis type 1..’ ‘has’ should be ‘have’

We have not modified the verb given the subject is “an increasing number” . Yet, please let us know you still have concerns about it and we can revise the sentence extensively.

  1. ‘The Neurofibromatosis type 1 (NF1) is an autosomal dominant condition with a birth incidence of approximately 1:2000-3000, caused by (mutations) in NF1,..’ -The preferred term is now ‘germline pathogenic variants’

We have modified the text according to the reviewer’s suggestion.

  1. ‘In this scenario of extreme phenotypic variability, there are some genotype-phenotype associations that we need to recognize because (strongly impacting) on genetic counseling and prognostication of the disease.’ - (strongly impacting)-rephrase to ‘they strongly impact’

We have rephrased the text.

  1. ‘The incidence is 1/2000-3000 individuals in the general population [1-3]’ -You must define ‘incidence’. Here you mean birth incidence. Please also cite source references rather than reviews so references 1 and 2 should be replaced.

We have now written “birth incidence” and have replaced references 1 and 2 with three, more appropriate references.

  1. ‘(In) About 50% of cases, it is caused by de novo mutations of NF1, on chromosome 17q11.2, encoding neurofibromin,..’ -English

We have corrected the sentence.

  1. ‘NF1 is characterized by highly variable inter- and intra-familiar expressivity and multi-systemic involvement. Clinical diagnosis has always been based on the clinical-radiological criteria defined in 1987 by the National Institutes of Health [8].’ -NOT always. How was it made before 1987? There are now new revised criteria that should be quoted so the authors should say since 1987 until 2021 and cite new reference. Although you do quote these later the use of ‘always’ here is inappropriate

We agree with the reviewer that this could be misunderstood. We have rephrased the sentence to clarify this point.

  1. ‘. CALMs may also be present in other clinical conditions, including LGSS, LEOPARD syndrome, piebaldism, McCune Albright syndrome, and Mismatch Repair Deficiency (MMR-D) syndrome’ The latter is usually referred to as Congenital’ in both that and McCune Albright syndrome the CALMS have a usually irregular border unlike NF1

We have added the word ‘constitutional’ for CMMRD syndrome. We have also specified the difference between CALMs in McCune Albright syndrome and in NF1.

  1. ‘Nevus anemicus (NA) is a congenital hypopigmentated skin abnormality’ -why is this anemic nevus in the table- be consistent

We apologize for any inconsistency. We have now added two references supporting our decision to include NA in Table 2.

  1. ‘In the last few years, a new ocular sign represented by choroidal anomalies (CAs) has been characterized in NF1 and included in the diagnostic finding [1].’ -is ref 1 really the most appropriate here?

We apologize. We have now removed this refence and have added three, more specific references.

  1. ‘OPGs are the most frequent tumors of the central nervous system (CNS) in children under the age of 7, with an incidence of 15-20% [19]’ -This is only in children who have been imaged. Over 66% of these are asymptomatic c and never become symptomatic. This figure needs context as MRI scanning of the head is not generally recommended in asymptomatic children with NF1

We thank the reviewer for pointing this out. We have now rephrased the entire sentence, underlining the fact that MRI is indicated for patients who are symptomatic or have ophthalmological signs.

  1. ‘However, almost half of the patients can develop clinical symptoms or signs (visual acuity deficit, pupillary pallor, strabismus, campimetric deficit, proptosis) [20]’ -I strongly disagree with this statement. It is NOT over half. That would mean >10% of NF1 children develop symptomatic OPG that is simply not true based on large epidemiological studies. https://www.ncbi.nlm.nih.gov/pmc/articles/PMC6849871/the paper you quote is a biased sample of mostly symptomatic OPG at diagnosis and not reflective of a total population or indeed of a ‘scanned’ population.

We are very grateful to the reviewer for their comment, and we totally agree. We have now changed the sentence and the reference.

  1. ‘With a 10% of possibility throughout life of evolving in MPNSTs, pNFs represent a high-risk condition in both morbidity and mortality [2,33]’ -Again the epidemiology is a bit suspect. It is by no means that all MPNST grow from plexiform tumors. The 10% figure is the lifetime risk in the entire population this might mean a higher than 10% risk in someone with a plexiform tumor

Again, we are grateful to the reviewer for their comment. We have now provided further details on this point.

  1. ‘MPNSTs are soft tissue sarcomas. In NF1, they have an incidence of about 8-16% and tend to occur earlier (20-40 years) respect to sporadic cases [34-36].’ -You need to define incidence. This is a lifetime risk. Please give an indication of what the childhood risk of MPNST is as that is the subject of the review

Unfortunately, we have not been able to find the data that the reviewer requested, also in light of the fact that diagnosis is usually made in adults. Although we are aware that MPNSTs can occur earlier (around 18thies), and we have direct experience with a few of our affected patients, not enough robust data are available and worth reporting.

  1. Section on vascular ‘Hypertension is relatively common in NF1, with an estimated frequency of 16–19% depending on literature source [1]. It is generally "essential", with development at any age and increasing with age. ‘Moyamoya vasculopathy is instead the most frequent cerebral arterial disease in NF1’ -Please give a figure for childhood as this review is about pediatric aspects. Both hypertension and Moyamoya are mostly adult onset

Although hypertension usually develops in adults, secondary forms resulting from renal stenosis and mid-aortic syndrome do exist and need to be listed. We have now added two references about these associations. Unfortunately, reports are very rare, and the very few available papers focusing on hypertension in childhood are often biased. Regarding Moyamoya vasculopathy, we respectfully but strongly disagree. There are several papers well documenting that the mean age for developing MMS in NF1 falls in childhood, and there is abundant literature on this. As an example, please see PMID: 25443089 (“The mean age of diagnosis for NF1 and MMS was 2.7 ± 2.1 years (range, 1-6 years) and 11.4 ± 8.3 years (range, 3.5-23 years), respectively.”).

  1. ‘The pathogenesis of MMS, associated or not to NF1, is unclear, but certainly independent of the type of germline mutations in NF1 [60].’ -Define MMS please. This is a long section on a condition that is rare in childhood but the audience needs to know how rare

We have now modified this section and describe Moyamoya vasculopathy, which we believe is more correct in this context. But again, Moyamoya is not actually that rare at all in childhood, and there are many papers debating the usefulness of MRI screening, for example in children under the age of 4, in order to detect asymptomatic forms of MMS. (PMID: 25443089)

  1. ‘Excluding the correlation to renal artery stenosis, it is still debate(able) whether a real connection exist(s),’ If the prevalence is 15% then surely there has to be. Either 15% is wrong-most likely or there is a link as renal artery stenosis is rare

We have now removed this sentence.

  1. Table ;’ juvenile xanthogranuloma (JXG) Children recs: Physical examination If JXG present: palpation of ganglionic areas and complete blood count’ -This is controversial and NOT recommended by the AACR guidelines or more recent GENTURIS. You need to make this clear

We absolutely agree with the reviewer, and this is now highlighted twice in the revised manuscript. We have also cited the recently published GENTURIS guidelines.

  1. ‘To date, five genotype-phenotype correlations are known and summarized in Figure 5 and 6.’ -The variants should be properly described. What does Met1149 and Arg1276 mean?

We apologize for any lack of clarity. The legend to Figure 5 has now been modified to clarify that amino acid changes at specific positions are grouped according to the observed phenotypic features. As different changes are possible and are reported for the same amino acid, we originally only indicated each change  with a three-letter code and its position in the protein sequence. We have now modified the figure to include the full name of the amino acids. The same information was already included in the legend to Figure 6.

  1. ‘Furtherly, we very recently observed CA absence in our NF1 patients with the Arg1809 substitution [91]’ -better to spell out CA here

We have now spelled out CA as requested.

Reviewer 2 Report

This review is very interesting to approach the relevance of genotype-phenotype associations to improve the surveillance and stratification of NF1 patients, to address the evaluations and the invasiveness of them.

1) Could the authors better explain the definition of  high-risk and (on the other side) a mild phenotype and low risk? Is this sentence referred to the clinical manifestations? Or what?

2) Could they add some more sentences about the use of MRI at the age at which children don't need anlagesic procedure? They think it should be added in the guidelines ? 

Author Response

Manuscript revision, round #1

Reply to reviewer #2

We have carefully considered the comments of the reviewer and revised our manuscript according to their suggestions. Our detailed point-by-point reply is reported below and typed in italics.

This review is very interesting to approach the relevance of genotype-phenotype associations to improve the surveillance and stratification of NF1 patients, to address the evaluations and the invasiveness of them.

We are grateful to the reviewer for their positive comments.

  • Could the authors better explain the definition of high-risk and (on the other side) a mild phenotype and low risk? Is this sentence referred to the clinical manifestations? Or what?

We thank the reviewer for raising such an interesting question. Unfortunately, we have not been able to find a clear definition. In the scientific literature, ‘severe phenotype’ usually refers to patients with oncological manifestations, high burden of cutaneous neurofibromas, disfiguring pNF, intellectual impairment, and so on. ‘Mild phenotypes’ include those with a lower number of neurofibroma and lower or no risk of OPG or pNFs.

We have now added a few lines in the revised version to clarify this point.

  • Could they add some more sentences about the use of MRI at the age at which children don't need anlagesic procedure? They think it should be added in the guidelines?

Patients under the age of 7 usually require sedation; we have now added this information and referenced it. As far as the guidelines are concerned, the power of a screening tool lies in its ability to identify a disease that can be treated in a way that can substantially change its natural history. In our opinion, this is not the right place to discuss issues concerning the usefulness of brain MRI screening in NF1. However, we are convinced that the need for sedation and the current lack of a specific and effective therapy for OPG are the two major reasons why brain MRI is not routinely recommended.

Round 2

Reviewer 1 Report

The authors have extensively revised the manuscript in line with reviewer comments and is much improved

There is just one minor issue remaining although ref 1-4 have been changed 3 and 4 are not source references. There is a better Finnish ref instead of 4 and another epi study should replace 3.

Author Response

We thank reviewer for their suggestion and positive comment. We changed ref. 2 and 4 as requested.

Please note that all figures are original and made by us or pictures from personal sereie sof patients or MRI.

Wherease tables 1-3 are extracted from two papers [11,13] properly cited and both of which are Open Access. Thus reuse of portions or extracts from the articles in other works is permitted.